# Improved Adsorption of the Toxic Herbicide Diuron Using Activated Carbon Obtained from Residual Cassava Biomass (*Manihot esculenta*)

**DOI:** 10.3390/molecules27217574

**Published:** 2022-11-04

**Authors:** Jordana Georgin, Diana Pinto, Dison S. P. Franco, Matias Schadeck Netto, Joseane S. Lazarotto, Daniel G. Allasia, Rutineia Tassi, Luis F. O. Silva, Guilherme L. Dotto

**Affiliations:** 1Research Group on Adsorptive and Catalytic Process Engineering (ENGEPAC), Federal University of Santa Maria, Av. Roraima, 1000-7, Santa Maria 97105-900, RS, Brazil; 2Universidad De La Costa, Calle 58 # 55-66, Barranquilla 080002, Atlántico, Colombia; 3Graduate Program in Environmental Engineering, Federal University of Santa Maria, Santa Maria 97105-900, RS, Brazil

**Keywords:** adsorption, residue, herbicide, activated charcoal

## Abstract

The production and consumption of cassava (*Manihot esculenta*) occur in several places worldwide, producing large volumes of waste, mostly in the form of bark. This study sought to bring a new purpose to this biomass through producing activated carbon to use as an adsorbent to remove the herbicide Diuron from water. It was observed that the carbon contains the functional groups of methyl, carbonyl, and hydroxyl in a strongly amorphous structure. The activated carbon had a surface area of 613.7 m^2^ g^−1^, a pore volume of 0.337 cm^3^ g^−1^, and a pore diameter of 1.18 nm. The Freundlich model was found to best describe the experimental data. It was observed that an increase in temperature favored adsorption, reaching a maximum experimental capacity of 222 mg g^−1^ at 328 K. The thermodynamic parameters showed that the adsorption was spontaneous, favorable, and endothermic. The enthalpy of adsorption magnitude was consistent with physical adsorption. Equilibrium was attained within 120 min. The linear driving force (LDF) model provided a strong statistical match to the kinetic curves. Diffusivity (D_s_) and the model coefficient (K_LDF_) both increased with a rise in herbicide concentration. The adsorbent removed up to 68% of pollutants in a simulated effluent containing different herbicides. Activated carbon with zinc chloride (ZnCl2), produced from leftover cassava husks, was shown to be a viable alternative as an adsorbent for the treatment of effluents containing not only the herbicide Diuron but also a mixture of other herbicides.

## 1. Introduction

A striking aspect of modern agriculture is the often-indiscriminate use of pesticides. Its positive points, such as high production, contrast with the environmental damage and damage to living organisms caused by these highly toxic compounds. Negative effects can be seen in non-target organisms [1], agricultural workers, manufacturers, and rural residents in low-income countries [2,3]. Diuron (N-(3,4-dichlorophenyl)-N-dimethylurea) is among the most used herbicides in the world for selective weed control [4]. This compound is classified as carcinogenic to humans and is considered a highly toxic substance by the European Union [5]. Its damage depends on the host, exposure time, and concentration. Several studies have confirmed Diuron’s toxicity to the environment and humans [6,7,8]. Due to its widespread use, Diuron has already been detected in many diverse water bodies around the world [4]. Accumulation of the herbicide in the environment occurs through effluents generated in the manufacturing stage, in transport to agricultural planting areas, and mainly through leaching after direct application to the soil [9]. Chemical and biological mechanisms can rapidly dissipate the compound in soil [10]. However, due to its low molar mass and high solubility and mobility, it is often leached, concentrating in different water bodies [11].

Due to these characteristics, different technologies have been studied to remove this contaminant from water resources. Adsorption with activated carbon offers a straightforward application process and has the excellent benefit of using leftover biomass. [12]. In this sense, several materials have already been successfully applied in the removal of Diuron and other pesticides, such as wheat husk treated with H_2_SO_4_ [13], corn cob [14], palm trunk [15], wood composites [16], mushroom residues [17], *Physalis peruviana* residues [18], bamboo stalk [19], the bark of the forest species *Cedrella fissilis* [20], araçá fruit bark [21], kaki seeds [22], leaves [23], baobab seed husks [4,24], bottom ash [25], and carbon nanotubes [26].

Cassava (*Manihot esculenta*) is a nutrient-rich plant from South America, where its root is one of the main sources of carbohydrates [27]. The cassava industry generates a large amount of solid waste, among which is the husk that makes up 3% to 5% of the total weight of its root. About 1 million tons of cassava husks are produced annually in Brazil [28]. Despite the possibility of its use as animal feed [29], it is impossible to fully consume this residue, which often leads to problems if it is disposed of improperly. Cassava peels can be used as an adsorbent. Their adsorptive properties have been observed in some studies for the removal of contaminants such as fluorine [30], methylene blue [31], rhodamine B [32], antibiotics [33], heavy metals [28,34,35,36,37], norfloxacin [38], nitrate [39], dye mixture [40], and malachite green [41].

No studies were found in the literature analyzing this residue’s adsorptive potential for herbicides. Therefore, this study produced activated carbon with zinc chloride (ZnCl_2_) from cassava husks. The carbonaceous material was used in batch studies to remove the herbicide Diuron. First, different characterization techniques were applied to the original and carbonized material to identify the possible chemical and structural changes caused by the pyrolysis. Then, pH studies were conducted, and adsorbent dosage was defined. Next, kinetic and isothermal studies were carried out for later determination of the thermodynamic parameters. Finally, a simulated effluent (Diuron + Atrazine + 2,4-dichlorophenoxyacetic acid) was treated with about 1 g L^−1^ of the charcoal produced in this study.

## 2. Results and Discussion

### 2.1. Characterization of Carbon-Based Precursor and Adsorbent Material

First, the yield of coal obtained in the study was around 29%. The ash content was about 4.6% of the material, indicating that a large part of the biomass (70%) was eliminated or transformed during the pyrolysis step. It is worth noting that several studies in the literature using other plant biomasses, with zinc chloride (ZnCl_2_) in a 1:1 ratio, obtained similar results [20,42,43,44,45,46]. In the study by Oliveira et al. [47], using cassava bagasse to produce activated charcoal, the yield was 4.1 ± 0.8% using sodium hydroxide (NaOH) during the pyrolysis step.

The FTIR spectra were acquired in an effort to identify the primary functional groups present in the samples both before (CP) and after carbonization (CPAC), with the main bands identified by letters (Figure 1A). The wide band in the region of 3432 cm^−1^ (a), present in both samples but with lower intensity in the carbonaceous material, corresponds to the N-H or O-H vibration stretch [39]. The band in the 2925 cm^−1^ region (b), which remained only in the original material, is related to asymmetric CH stretching vibrations [44]. The band in the region of 1653 cm^−1^ (c), present in the two samples, corresponds to the elongation of the carboxylate bonds [17]. The small band in the 1378 cm^−1^ region (d), present only in the CP sample, is attributable to δ-O–H and ν-C=O vibrations [48]. However, the band in the region of 1022 cm^−1^ (e), present in both materials but with lower intensity in the CPAC, may be related to the CN or CO stretching vibration [49]. The final band at 603 cm^−1^ (f), present in both materials, is attributed to the ν O–H bond [48]. A general analysis shows that although most bands remained in the material after pyrolysis, most of them lost intensity. The primary functional groups were methyl, carbonyl, and hydroxyl, which are structures related to the byproducts of lignin and cellulose degradation [50,51,52]. This was expected given the origin of the bark. It should be noted that the permanence of functional groups in the activated carbon sample is necessary for the adsorption process, as they correspond to active sites of attraction of the adsorbate molecules [53].

Figure 1B shows the XRD diffractograms of the samples, which give more information about the structure of the materials. The two non-crystalline peaks in the CP sample were a result of crystalline cellulose (II). In addition, the presence of additional organic matter, such as lignin and hemicellulose, was responsible for the peak width [54,55,56]. The broad amorphous peak persisted after the ZnCl2 carbonization stage. However, a new peak also appeared in the 002 planes of the graphite crystal structure produced during the pyrolysis process [44]. Therefore, it was confirmed that carbon in its amorphous form was the predominant phase in the materials. This indicates the presence of unorganized structures, which is a characteristic that may favor the adsorption of Diuron since the possible empty spaces could accommodate the adsorbate molecules. In the literature, other coals from different plant biomasses have been found with similar patterns, such as those from peanut husk [57,58], Queen palm fruit endocarp [45], *Cedrella fissilis* husk [20], araucaria bark [59], argan bark [60] and Indonesian Kesambi wood [61].

Scanning electron microscopy was performed to analyze the material surface before (Figure 2A–C) and after carbonization with ZnCl_2_ (Figure 2D–F). The surface of the original material was highly heterogeneous, containing cracks and rounded geometries that corresponded to the starch on the surface [62]. Notably, the starch spheres were consumed during the carbonization process, which could be explained by the amylase not being bound to the cellulose matrix. Besides that, the carbonization process changed the morphology significantly, and new spaces emerged. In addition, irregularities and protrusions gained new shapes, making the surface less smooth and irregular. Finally, all the particles corresponding to the starch disappeared, confirming that the carbonization was efficient, and the micrographs showed void spaces that could be used to accommodate Diuron molecules.

The developed activated carbon had a surface area of 613.7 m^2^ g^−1^. When analyzing the classification of N_2_ adsorption–desorption isotherms (Figure 3A), it was observed that they are type I [63]. This type of isotherm is related to microporous materials [64]. A hysteresis loop of type H_4_ was also observed, which extended from 0.6 to 1. According to the IUPAC, this loop is associated with narrow slit-shaped pores [65]. The carbonaceous material showed significant pore development (Figure 3B), presenting a pore diameter of 1.126 nm and pore volume of 0.337 cm^3^ g^−1^. This value corresponded to the large presence of micropores (<2 nm), in agreement with the hysteresis loop in the isotherms. These results reflect the effectiveness of the synthetic method employed. ZnCl_2_ behaves as a chemical activating agent and induces the formation of pores [66].

### 2.2. Equilibrium Isotherms

Isotherm curves were constructed at four different temperatures (Figure 4). These studies showed the interaction mechanisms between the adsorbent and the adsorbate and allowed the estimation of thermodynamic parameters. The curves represent the function of the amount of Diuron present on the surface of the CPAC (q_e_, mg g^−1^) based on how much herbicide was present in the aqueous medium (C_e_). The temperature varied from 298 to 328 K in different herbicide concentrations. Both curves showed similar and favorable behavior for the adsorption of Diuron, regardless of temperature. However, the shape of a plateau was not observed, which may indicate that there are still places available for the adsorption of Diuron molecules. Similar behavior was observed throughout the 2,4-D herbicide and activated carbon adsorption processes [17].

It should be noted that these types of curves can be classified as L1 [67]. The increase in temperature favored the adsorption capacity, which ranged from 164 to 222 mg g^−1^, analyzed at the highest concentration (200 mg g^−1^). The increase in concentration also caused an increase in capacity, ranging from 113 to 222 mg g^−1^, for a concentration range of 50 to 200 mg L^−1^, at the optimum temperature (328 K). When using activated carbon to remove Diuron, the most effective temperature was found to be 318 K, with an adsorption capacity of 485.11 mg g^−1^ [68]. At 318 K, graphene oxide that had been coated with iron oxide nanoparticles performed better as an adsorbent (30.29 mg g^−1^) in the adsorption of Diuron [69]. The increase of the adsorption capacity with temperature can be attributed to different effects, such as: (i) increased solubility, facilitating the movement of the adsorbate and allowing it to reach new sites, and (ii) modification of the adsorbent structure, causing it to expand and opening regions inside the material that were not previously available [59].

Table 1 provides the parameters that fit the Langmuir, Freundlich, and Dubinin–Radushkevich models. First, when analyzing only the statistical adjustments for the coefficients R^2^ and R^2^_adj_, it was noticed that the highest values were obtained by the Freundlich isotherm (>0.9669), followed by the Langmuir isotherm (>0.9081), and the Dubinin–Radushkevich (>0.8933). When analyzing the values of ARE and MSR, the lowest values obtained were for the Freundlich model, with ARE < 6.031 and MSR < 136.7, whereas the Langmuir model obtained values of ARE < 8.204 and MSR < 431.1. Finally, the least satisfactory fits were obtained for the Dubinin–Radushkevich model, with ARE < 8.304 and MSR < 483.8, confirming the best fit for the Freundlich model. This model presents difficult parameters to correlate to the adsorption isothermal phenomena overall. It is worth mentioning that the adsorption capacities obtained by the Langmuir model were highly consistent with the values obtained experimentally, increasing from 150 to 199 mg g^−1^ with increasing temperature in the system. In this study, the Langmuir values were considered to estimate the thermodynamic data, where the value of K_F_ was ((mg g^−1^) (mg L^−1^)^−1^), and the equilibrium constant was not considered [70].

Table 2 provides information on studies described in the literature on Diuron removal. The objective is to briefly compare the CPAC adsorbent with the others already described. It is noted that CPAC has the fourth-best adsorption capacity (222 mg g^−1^). Although the concentration variations in the studies are quite significant, it can be said that the application of residual cassava husk presents real potential for producing activated carbon for subsequent use as an adsorbent in removing Diuron.

### 2.3. Thermodynamic Studies

The thermodynamic parameters investigated in the adsorption of Diuron in CPAC are listed in Table 3. The thermodynamic parameters make it possible to identify the nature and spontaneity of the adsorption process.

The K_e_ values increased from 1.911 to 2.689 as the system temperature increased from 298 to 328 K, indicating the favoring of the process at the highest temperature. The Gibbs energy increased negatively from −41.52 to −46.63 kJ mol^−1^, with the most negative values found at 328 K, demonstrating that the adsorption of Diuron in the CPAC was spontaneous and favorable. On the other hand, the enthalpy value (∆H^0^) remained positive (11.47 kJ mol^−1^), in agreement with the isothermal studies, thus confirming an endothermic process. The magnitude of ΔH^0^ follows strong physical forces, which may be electrostatic attraction [77]. Thus, it is possible to infer that in this study, the predominant process was physisorption, which may be reversible, allowing desorption. Furthermore, the positive value of ΔS^0^ (0.0259 kJ mol^−1^ K^−1^) confirms a high affinity for Diuron molecules on the surface of CPAC. Although most studies reporting the adsorption of Diuron did not analyze the influence of temperature on the adsorption (with most studies being carried out at room temperature, as reported in Table 2), the studies by Bahri et al. [78] and Barbosa de Andrade et al. [69] also observed an endothermic behavior in herbicide removal.

### 2.4. Diuron Adsorption Kinetics

Kinetic studies were performed using different concentrations of the adsorbate. For this study, the time was varied from 0 to 180 min (t) using the adsorption capacity (q) for concentrations of 50, 100, and 200 mg L^−1^. The kinetic curve profiles are illustrated in Figure 5.

First, it was observed that the adsorption capacity increased independently of the adsorbate concentration. On the other hand, the system reached equilibrium a little faster at the lowest concentration studied and a little slower at the other concentrations. Therefore, for the concentration of 50 mg L^−1^, the system reached equilibrium after around 30 min with a maximum capacity of 90 mg g^−1^. For concentrations of 100 and 200 mg L^−1^, equilibrium was only reached after 120 min, with capacities of 133 and 165 mg g^−1^ for concentrations of 100 and 200, respectively. Despite the slight variance, it can be concluded that the CPAC has a fast kinetic rate, which is advantageous for full-scale applications. A high volume of polluted effluent can be treated in a shorter amount of time, resulting in less energy expenditure. The CPAC was also observed to have a rapid rate of initial adsorption followed by a slower rate as the adsorption approached the equilibrium time; this behavior is expected since, in the first minutes, most of the adsorption sites on the surface of the CPAC were free, whereas the rate tended to slow down as it approached saturation [79].

The linear driving force model (LDFM) showed a good fit for the adsorption kinetic data of the CPAC/Diuron system, as shown in Table 4. The R^2^ value was above 0.9529, and the ARE and MSE values were below 14% and 165, respectively (mg g^−1^)^2^. The predicted adsorption values were in agreement with the experimental ones: 96, 135, and 157 mg g^−1^, for concentrations of 50, 100, and 200 mg L^−1^, respectively. It should be noted that the adsorption rate increased as the concentration increased, due to the differences between the initial concentration of Diuron and the concentration on the surface of the adsorbent [80]. In this sense, it was observed that the values of k_LDF_ increased from 0.74 to 1.35 × 10^−3^ s^−1^ with increasing concentration. Showing similar behavior, the diffusivity values also increased from 0.77 to 1.41 × 10^−8^ cm^2^ s^−1^ with increasing adsorbate concentration. Georgin et al. [44] also showed similar behavior of diffusivity and k_LDF_ in the adsorption of ketoprofen with activated carbon.

### 2.5. Proposal of Adsorption Mechanism

To propose an adsorption mechanism it is necessary to consider the results obtained from the FTIR, the pH_pzc_ (6.5, according to the Appendix A), the pK_a_ of diuron, and the magnitude of the standard enthalpy change. The FTIR analysis indicates the presence of classical groups in activated carbon structures, which consist of C-C, C=C, C-O, and -OH, with the presence of aromatic rings. The pH_pzc_ gives the charge at the surface of the adsorbent at the chosen pH, in this case, 7, which indicates that for values above 6.5 the surface of the CPAC is mainly negative. The diuron molecule has a high pK_a_ value (around 13.2) which leads to diuron being 100% in neutral form at pH 7. From the magnitude of the ΔH^0^ (~11 kJ mol^−1^) it is possible to consider that the adsorption mechanism is due to physical interaction, without the presence of chemical bonding/electron exchange. Employing all these results and the possibility of hydrogen bonding, electrostatic interaction, π–π interactions, and anion-π interaction, the adsorption mechanism is proposed as depicted in Figure 6.

### 2.6. Adsorption Efficiency against a Simulated Effluent

The adsorbent showed strong removal efficiency in a mixture of herbicides, obtaining the removal of 68.65%, as shown in Figure 7. This performance is close to, or even above other studies found in the literature for simulated effluents containing different herbicides. When using peanut skin treated with H_2_SO_4_ to remove an effluent containing 2,4-D and Atrazine, the authors observed removals of 15.01%, 46.0%, and 72.02% for the dosages of 1, 3, and 5 g L^−1^, respectively [81]. In another study, residual husks of wheat Fagopyrum esculentum treated with H_2_SO_4_ were used to treat samples of water from real rivers contaminated with 2.4-D [13]. This study observed the removal of around 76% for both analyzed rivers. Finally, the mushroom residues (*Agaricus bisporus*) were carbonized and used to treat a river contaminated with 2.4-D, where the removal of 70% was observed [17].

## 3. Materials and Methods

### 3.1. Utilized Chemicals and Reagents

The chemical compounds Diuron 3-3,4-dichlorophenyl-1,1-dimethylurea (chemical equation: C_9_H_10_Cl_2_N_2_O; M_w_: 233.1 g mol^−1^; M_V_: 170.1 cm^3^ mol^−1^; pK_a_: 3.7; solubility value of diuron in water is 42 mg L^−1^ at 298 K) and zinc chloride (ZnCl_2_) [82,83], were purchased from Sigma-Aldrich USA (all analytical grade). Diuron has a limited water solubility, thus the stock solution was made by precisely dissolving 1 g L^−1^ of this herbicide in methanol to obtain a concentration of 1000 mg L^−1^. The stock solution was further diluted to obtain the desired concentrations. The adsorption experiments were performed at the pH_natural_ of the solution (pH_natural_ = 7) with an adsorbent dosage of 0.5 g L^−1^. The experiments and solution formulations were conducted exclusively with deionized water.

### 3.2. Obtaining and Characterizing the Precursor Material and Activated Carbon

Cassava is a food traditionally consumed in Brazil, therefore the residual husks were obtained from rural producers located in the South of Brazil, specifically in the state of Rio Grande do Sul. The methodology employed for the fabrication of the activated carbon is described in the Appendix A. In short, the mixture of 25 g of ZnCl_2_ and cassava husk was pyrolyzed at 923.15 K for 80 min, after which the activated carbon was washed using HCl (10 mol L^−1^) until the water reached pH = 7. The activated carbon, entitled cassava husk activated carbon (CHAC), was sieved, dried, and separated to be further employed. The characterization methodology is shown in the Appendix A.

### 3.3. Diuron Adsorption Experiments

For the adsorption assays, a thermostatic stirrer (MA093, Marconi, São Paulo, Brazil) was used at 160 rpm during all experiments. The content of Diuron in the aqueous medium was measured using a spectrophotometer (UV mini 1240, Shimadzu, Tokio, Japan) at the wavelength of maximal Diuron absorption (248 nm). The studies were carried out in triplicate (n = 3), and the Diuron/CPAC separation was performed on the materials by centrifuging them (Centribio, 80-2B, São Paulo, Brazil) at 4000 rpm for 25 min after each experiment. All kinetic and isothermal studies described below were performed at the pH_natural_ (pH = 7) and with a fixed dosage of activated carbon of 0.5 g L^−1^. In order to achieve the equilibrium curves, temperatures of 298, 308, 318, and 328 K were used. Diuron starting concentrations of 0, 50, 100, 150, and 200 mg L^−1^ were supplied in 25 mL Erlenmeyer flasks. The samples were shaken for 5 h, ensuring equilibrium in the Diuron/CPAC system. Finally, the kinetic studies were carried out by varying the initial concentrations of Diuron at 50, 100, and 200 mg L^−1^ added to 25 mL Erlenmeyer flasks of solution at a room temperature of approximately 298 K, obtaining samples with the aid of an aliquot of 5 mL at different time intervals ranging from 0 to 180 min. The determination of adsorption capacities (at any time and at equilibrium, mg g^−1^) and percentage of removal (R, %) was conducted according to the batch adsorption equations shown in the Appendix A.

### 3.4. Equilibrium Models and Thermodynamic Parameters

Freundlich [84], Dubinin–Radushkevich [85], and Langmuir [86] models were adopted to be fitted to the experimental isothermal data. The thermodynamic constants (Gibbs free energy, enthalpy, and entropy) were based on the equilibrium constant of the best-fitted isotherm. In this case, the methodology was proposed by Tran [87]. Detailed information regarding the isotherm models and the estimation of thermodynamic parameters can be found in the Appendix A.

### 3.5. Diuron Adsorption Kinetics

The kinetics were mainly focused on the effect of adsorbate initial concentration on the time to reach equilibrium and the possible adsorption mechanism. In this work, the linear driving force (LDF) model was selected [88]. Considering that the Freundlich model presented the best fit to the experimental data, the LDF is shown in Equation (1) (the deduction of the model can be found in the Appendix A):(1)$$\frac{d\bar{q}}{\mathrm{dt}}{=k}_{\mathrm{LDF}}\left(K_{F}{\left(C_{0}{-D}_{0}\bar{q}\right)}^{1/n}-\bar{q}\right)$$
(2)$$\bar{q}\left(t=0\right)=0$$
where k_LDF_ is the LDF parameter (s^−1^), K_F_ is the Freundlich parameter ((mg g^−1^) (mg L^−1^)^−1/n^), 1/n (dimensionless) is the heterogeneity parameter, C_0_ is the initial concentration (mg L^−1^), D_0_ is the adsorbent dosage (g L^−1^), q is the adsorption capacity (mg g^−1^), and t is the time (s).

### 3.6. Model Fitting, Differential Equation Solution, and Model Quality Fit

The parameter estimation, equation solution, and model evaluation were carried out through MATLAB scripting. Built-in functions were employed accordingly: *particleswarm* was used for the determination of the parameter’s initial guess, *nlinfit* was employed for the determination of the model parameter without any constraints, *lsqnonlin* was employed for the determination of the parameter with constraints, and *ode15s* was the solver employed for the solution of the LDF model. The quality of the models’ fit was found by employing the statistical parameters: The equations for each statistical indicator are presented in the Appendix A and include the determination coefficient (R^2^), adjusted coefficient of determination (R^2^_adj_), average relative error (ARE, %), and minimal squared error (MSE, (mg g^−1^)^2^).

### 3.7. Adsorption Performance in a Simulated Effluent Containing Diuron

During the growth of crops, the number of pesticides used varies greatly. This is because many are efficient only for certain weed species or even for certain leaf formats. Therefore, it is likely that a mixture of products is leached into water resources because, despite being applied at different times, they have a long shelf life and are highly persistent. Because of this, a mixture was prepared to contain not only the herbicide Diuron but also Atrazine, which is an herbicide used to control broadleaf weeds and grasses [21], and 2,4-dichlorophenoxyacetic acid (2,4-D), which is also used to control weeds [17]. The mixture was prepared in 100 mL of deionized water containing Diuron at a concentration of 50 mg L^−1^, and Atrazine and 2,4-D at 10 mg L^−1^ each. Then, a dosage of 1 g L^−1^ of the charcoal prepared in the study was added to the mixture, and the pH was measured. The dosage was increased because the experimental dosage (0.5 g L^−1^) did not achieve satisfactory removal. The sample was stirred for 4 h at room temperature (298 K), and the natural pH of the solution was 6.1 under stirring at 160 rpm. After the adsorption process, the sample was collected and scanned using the Shimadzu UV-2600 spectrum. The area under the curve of each spectrum was used to determine of the percentage of removal using the Origin pro 2016 software.

## 4. Conclusions

Residual husks from the cassava production chain were successfully carbonized using zinc chloride (ZnCl_2_) as an activating agent. The carbonaceous material used to remove the Diuron had a surface area of 613.7 m^2^ g^−1^ and a pore diameter of 1.178 nm. The amorphous structure remained in the material. However, the main presence of carbon after the pyrolysis proves that the carbonization was effective. The carbonization developed several new voids spread over the entire surface of the sample. Good adsorption capacity results were obtained at the dosage of 0.5 g L^−1^ and under natural pH conditions. An increase in temperature in the system favored the adsorption capacity. The maximum capacity (222 mg g^−1^) was obtained at 328 K. The Freundlich isotherm model presented the best statistical adjustment to the equilibrium data. The thermodynamic parameters confirmed a process of physical and endothermic nature (∆H^0^ = 11.47 kJ mol^−1^). The adsorption kinetic profiles indicated that the process reached equilibrium quickly, mainly at the lowest concentration. The linear driving force (LDF) model provided a good representation of the kinetic curves. In a sample of water tainted with several herbicides, the adsorbent worked effectively, eliminating roughly 68 percent of the pollutants.

## Figures and Tables

**Figure 1 molecules-27-07574-f001:**
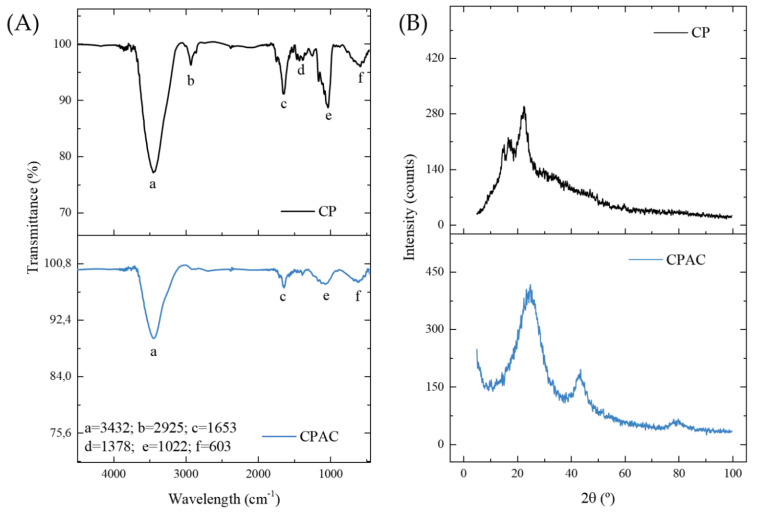
FTIR spectra (**A**) and XRD (**B**) patterns for CP and CPAC samples.

**Figure 2 molecules-27-07574-f002:**
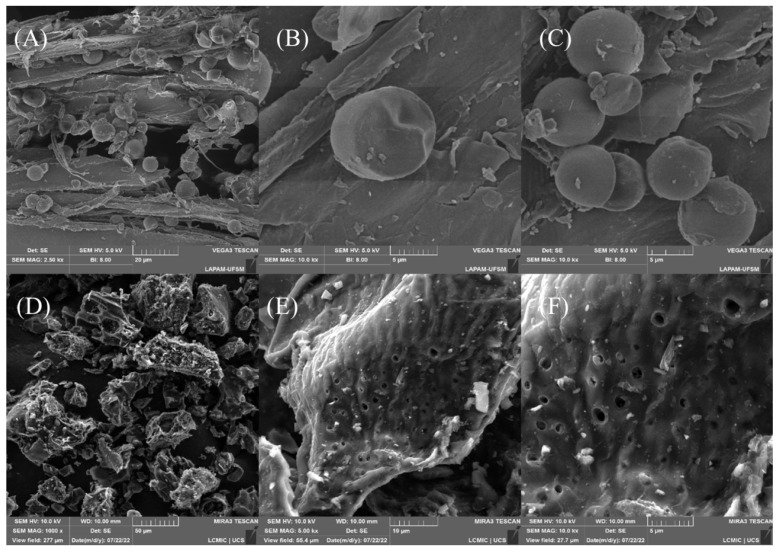
Micrographs of the material surface before (**A**–**C**) and after carbonization (**D**–**F**).

**Figure 3 molecules-27-07574-f003:**
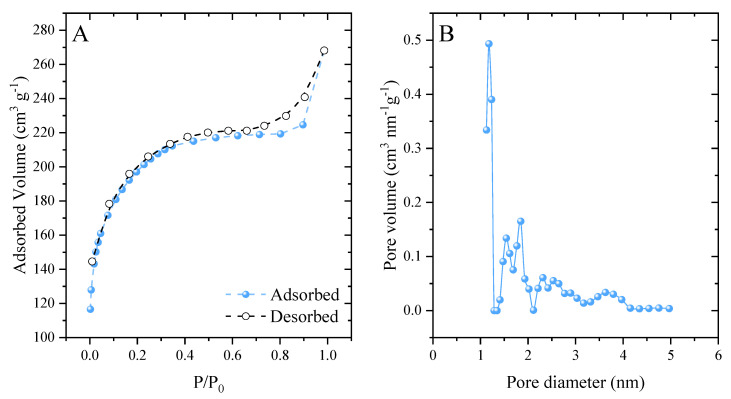
N_2_ ads/des isotherms (**A**) and pore size distribution (**B**).

**Figure 4 molecules-27-07574-f004:**
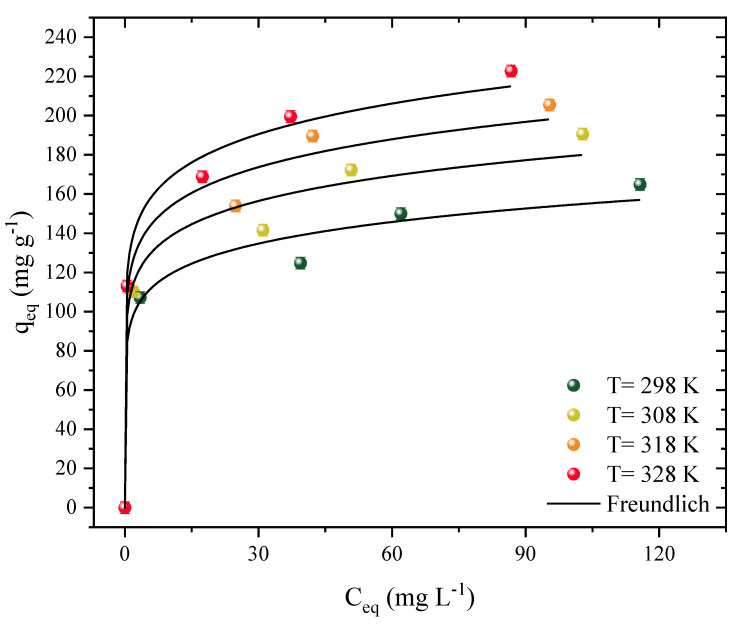
Adsorption isotherm data and Freundlich model prediction of Diuron in CPAC (D_0_ = 0.5 g L^−1^, pH_natural_ = 7, V = 25 mL).

**Figure 5 molecules-27-07574-f005:**
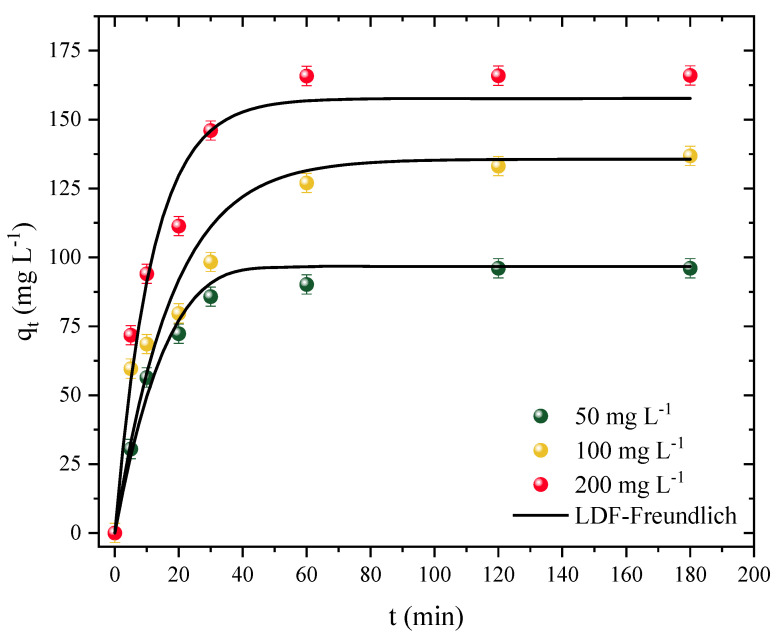
Diuron adsorption kinetic curves in CPAC (T = 298 K, D_0_ = 0.5 g L^−1^, pH_natural_ = 7, V = 25 mL).

**Figure 6 molecules-27-07574-f006:**
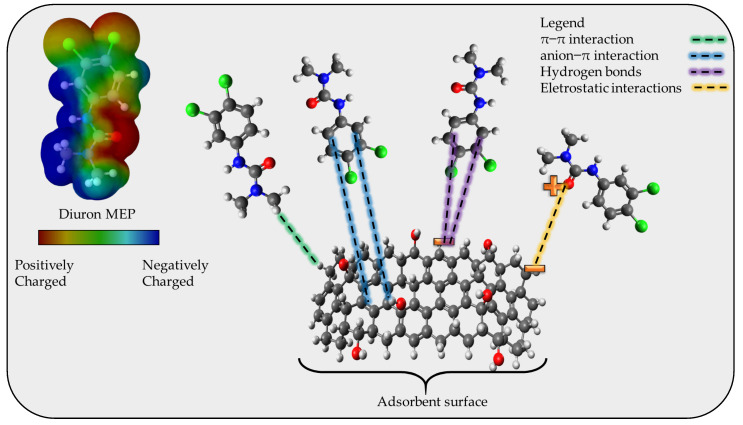
Proposed diuron adsorption mechanism onto the CPAC.

**Figure 7 molecules-27-07574-f007:**
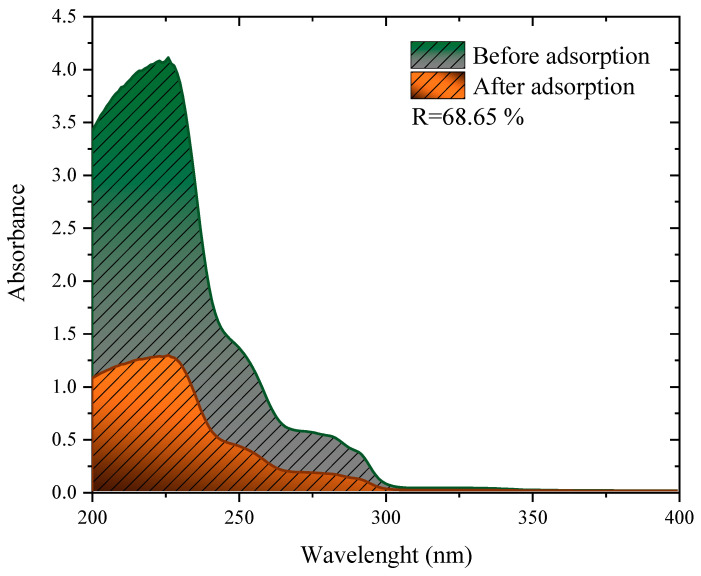
Invisible spectra of the simulated effluent before and after adsorption with 1 g L^−1^ of charcoal from the cassava husk.

**Table 1 molecules-27-07574-t001:** The isothermal parameters for Diuron adsorption on CPAC.

Temperature (K)				
Model	298	308	318	328
Langmuir				
q_mL_ (mg g^−1^)	150.9	172.9	186.0	199.8
K_L_ (L mg^−1^)	0.6685	0.8505	1.783	2.705
R^2^	0.9593	0.9546	0.9540	0.9590
R^2^_adj_	0.9185	0.9091	0.9081	0.9180
ARE (%)	7.500	8.035	8.204	7.240
MSR (mg g^−1^)^2^	229.9	342.8	413.7	431.1
Freundlich	298	308	318	328
K_F_ ((mg g^−1^)(mg L^−1^)^−1/nF^)	91.5	106.3	118.0	129.5
1/n_F_ (dimensionless)	0.1133	0.1134	0.1135	0.1136
R^2^	0.9834	0.9819	0.9851	0.9931
R^2^_adj_	0.9669	0.9638	0.9702	0.9863
ARE (%)	5.100	6.031	5.379	4.168
MSR (mg g^−1^)^2^	93.49	136.7	134.3	72.09
Dubinin–Radushkevich	298	308	318	328
q_mDR_ (mg g^−1^)	146.8	168.4	183.1	197.2
Β × 10^7^ (kJ^2^ mol^−1^)	7.856	3.754	1.108	0.5686
R^2^	0.9524	0.9467	0.9489	0.9540
R^2^_adj_	0.9048	0.8933	0.8978	0.9080
ARE (%)	7.689	8.207	8.304	7.359
MSR (mg g^−1^)^2^	268.7	402.5	459.9	483.8

**Table 2 molecules-27-07574-t002:** Adsorbents developed and used for removing the pesticide Diuron from aqueous media: parameters and maximum adsorption capacities compared to CPAC.

Adsorbent	T (K)	C_0_	Isotherm Model	q_m_ (mg g^−1^)	Reference
CPAC	328	50–200	Freundlich	222	This work
AC from baobab seeds hulls	303	5–20	Langmuir	65.7	[4]
Bottom ash	313	20	Langmuir	349.52	[71]
Natural Fibers from Waste African Baobab	298		Langmuir	400	[24]
Poly (methacrylic acid) PMMA	298	100	Langmuir	14.58	[72]
Poly (acrylic acid) PAA	298	100	Langmuir	7.32	[72]
Activated carbon	318	13–38	Langmuir	485.11	[68]
Graphene oxide decorated with iron oxide nanoparticles	318	10	Langmuir	30.29	[69]
Multiwalled carbon nanotubes	298	1.13–9.07	Langmuir	48.60	[73]
Carbon nanotubes synthesized from plastic waste	303	5–25	Hill	40.37	[74]
*Trametes versicolor* immobilized on pinewood	298	0.05	Langmuir	0.610	[75]
Commercial organophilic clay	308	5–20	Langmuir/Freundlich	56.49	[76]
Carbon nanotubes	298		Polanyi–Manes	182	[26]

**Table 3 molecules-27-07574-t003:** Thermodynamic parameters for adsorption of Diuron in CPAC.

T(K)	K_e_ × 10^−7^	ΔG^0^ (kJ mol^−1^)	ΔH^0^ (kJ mol^−1^)	ΔS^0^ (kJ mol^−1^ K^−1^)
298.1	1.911	−41.52	11.47	0.0259
308.1	2.216	−43.29		
318.1	2.456	−44.97		
328.1	2.689	−46.63		

**Table 4 molecules-27-07574-t004:** The kinetic parameters were estimated for the adsorption of Diuron in the CPAC.

Model	Diuron Concentration (mg L^−1^)		
	50	100	200
LDF-Freundlich			
q_pred_ (mg g^−1^)	96.74	135.6	157.7
k_LDF_ × 10^3^ (s^−1^)	0.7486	0.8105	1.357
D_S_ × 10^8^ (cm^2^ s^−1^)	0.7798	0.8443	1.414
R^2^	0.9833	0.9529	0.9654
ARE (%)	6.003	14.548	9.077
MSE (mg g^−1^)^2^	20.37	165.7	119.9
q_exp_ (mg g^−1^)	96.05	136.8	166.0

## Data Availability

The data will be available on when requested.

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
