# Peer review of "Improved Adsorption of the Toxic Herbicide Diuron Using Activated Carbon Obtained from Residual Cassava Biomass (Manihot esculenta)"

_molecules, 2022, doi:10.3390/molecules27217574_

Round 1

Reviewer 1 Report

I think that the article can be published in this form.

The article presents understandable figures and tables that give an idea of ​​the study.

An electron scanning microscope was used in the work, which made it possible to study the surface before and after carbonization.

Previously, the effect of temperature on adsorption processes involving diuron has not been studied. In this work, we studied 4 temperatures, which made it possible to elucidate the mechanism of interaction between the adsorbent and the adsorbate.

It follows from the work that the adsorbent proposed by the authors effectively removes pollutants (68%).

I have been treating water from highly radioactive waste for many years. This is a slightly different area.

I believe that this article will be of interest to specialists working in the field of ecology.

Author Response

Dear reviewer, thank you for the compliments. The authors are grateful for the approval.

Reviewer 2 Report

The paper on “Improved adsorption of the toxic herbicide Diuron using activated carbon obtained from residual cassava biomass (Manihot esculenta)” by Georgin et al. is an interesting study. Though the work is nicely presented, however, here are few comments which must be addressed:

1. What is the solubility of Diuron, readers will be interested in knowing it.

2. Authors didn’t mention the particle size of activated carbon after sieving.

3. In which concentration range experiments were performed, clearly state what initial concentration of the adsorbate was used for obtaining the adsorption isotherms

4. It is not clear whether authors carbonized or activated the precursor, clarify.

5. In SEM Fig 3c authors are showing some microspheres but no discussion is made and what happened after transformation of the material.

6. ZPC and Boehm titration may be included too to characterize the material as it will shed a light on its adsorption efficiency and authors can correlate it with FTIR too.

7. Discussion on the mechanism of the adsorbent-adsorbate needs improvement.

Author Response

1 What is the solubility of Diuron, readers will be interested in knowing it.

Response: Dear reviewer, the solubility value of diuron in water is 42 mg L-1 at 298 K (RODRIGUES & ALMEIDA, 2011), the same was added in the article.

https://doi.org/10.1590/S0103-84782013001100007

2  Authors didn’t mention the particle size of activated carbon after sieving.

Response: Dear reviewer, the value is mentioned in the supplementary material.

3 In which concentration range experiments were performed, clearly state what initial concentration of the adsorbate was used for obtaining the adsorption isotherms

Response: Dear reviewer, the requested information appears on line 107.

4 It is not clear whether authors carbonized or activated the precursor, clarify.

Response: Dear reviewer, the activation and pyrolysis process occurs simultaneously, as the carbon is carbonized and activated through ZnCl2.

5 In SEM Fig 3c authors are showing some microspheres but no discussion is made and what happened after transformation of the material.

Response: Dear reviewer, the authors added a discussion regarding the starch spheres as follows:

“Notably, the starch spheres are consumed due to the carbonization process, the possible explanation for this is the amylase not being bounding to the cellulose matrix”

6 ZPC and Boehm titration may be included too to characterize the material as it will shed a light on its adsorption efficiency and authors can correlate it with FTIR too.

Response: Dear reviewer, the pHpzc was added to the supplementary material and mentioned in the adsorption mechanism section. However, Boehm's titration will not be possible to be conducted at the current time.

7 Discussion on the mechanism of the adsorbent-adsorbate needs improvement.

Response: The following section was added to the manuscript with the objective of elucidating the adsorption mechanism interaction.

3.5 Proposal of adsorption mechanism

To propose an adsorption mechanism is necessary to consider the results obtained from the FT-IR, the pHpzc (6.5, according to the Supplementary material S.7), the pKa of diuron, and the magnitude of the standard enthalpy change. The FT-IR analysis indicates the presence of classical groups in activated carbon structures, which consist of C-C, C=C, C-O, and –OH, with the presence of aromatic rings. The pHpzc indicates what is the charge at the surface of the adsorbent at chosen pH, in this case, 7, this indicates that for values above 6.5 the surface of the CPAC is mainly negative. The diuron molecule has a high pKa value (?) which leads to being 100% in neutral form. From the magnitude of the ΔH0 (~11 kJ mol-1) it is possible to consider that the adsorption mechanism is due to physical interaction, without the presence of chemical bonding/electron exchange. Employing all these results and the possibility of hydrogen bonding, electrostatic interaction, π-π interactions, and anion-π interaction, the adsorption mechanism was proposed as depicted in Figure 7.

Reviewer 3 Report

1. Need to reformat the equations.

2. It would be better if the graphs in Fig. 1 and Fig. 2, is combined in one figure (one for FTIR and one for XRD) to see the comparison between CP and CPAC clearly.

3. First person point of view is considered informal, and thus, it is not encouraged in academic writing. Authors should refrain from using 'We'.

4. Need to elaborate more on why the reaction favors high temperature (328K) rather than room/lower temperature. Current discussion is too general.

Author Response

8 Need to reformat the equations.

Response: Dear reviewer, the equations in the manuscript have the font exchanged and centralized according to the text. According to the authors' guide for the Molecules Journal, the only need is that the equations should be able to be edited, which is in accord as well.

9 It would be better if the graphs in Fig. 1 and Fig. 2, is combined in one figure (one for FTIR and one for XRD) to see the comparison between CP and CPAC clearly.

Response: Dear reviewer, thank you for the suggestion, Figures 1 and 2 have been merged as one.

Figure 1. FT-IR spectra (A) and XRD patterns for CP and CPAC samples.

10 First person point of view is considered informal, and thus, it is not encouraged in academic writing. Authors should refrain from using 'We'.

Response: Dear reviewer, thank you for the comment. The authors agree and understand your point of view, the usage of “We” was a miss translation.

11 Need to elaborate more on why the reaction favors high temperature (328K) rather than room/lower temperature. The current discussion is too general.

Response: Dear reviewer, the following phrase was added to the manuscript

The increase of the adsorption capacity with the temperature can be attributed to different effects such as i) solubility increase, facilitating the movement of the adsorbate thus allowing it to reach new available sites; ii) modification of the adsorbent structure, in some cases, the adsorbent expand and further opens regions inside the material that were not previously available [66].